# The Role of NADPH Oxidases and Oxidative Stress in Neurodegenerative Disorders

**DOI:** 10.3390/ijms19123824

**Published:** 2018-11-30

**Authors:** Anuradha Tarafdar, Giordano Pula

**Affiliations:** Institute of Biomedical and Clinical Sciences (IBCS), College of Medicine and Health (CMH), St. Luke’s Campus, University of Exeter, Exeter EX4 4QL, UK; G.Pula@exeter.ac.uk

**Keywords:** NADPH oxidases, reactive oxygen species, dementia, Alzheimer’s disease, amyloid beta, cerebral amyloid angiopathy

## Abstract

For a number of years, nicotinamide adenine dinucleotide phosphate (NADPH) oxidases (NOX) was synonymous with NOX2/gp91^phox^ and was considered to be a peculiarity of professional phagocytic cells. Over the last decade, several more homologs have been identified and based on current research, the NOX family consists of NOX1, NOX2, NOX3, NOX4, NOX5, DUOX1 and DUOX2 enzymes. NOXs are electron transporting membrane proteins that are responsible for reactive oxygen species (ROS) generation—primarily superoxide anion (O_2_^●−^), although hydrogen peroxide (H_2_O_2_) can also be generated. Elevated ROS leads to oxidative stress (OS), which has been associated with a myriad of inflammatory and degenerative pathologies. Interestingly, OS is also the commonality in the pathophysiology of neurodegenerative disorders, such as Alzheimer’s disease (AD), Parkinson’s disease (PD), Huntington’s disease (HD), amyotrophic lateral sclerosis (ALS) and multiple sclerosis (MS). NOX enzymes are expressed in neurons, glial cells and cerebrovascular endothelial cells. NOX-mediated OS is identified as one of the main causes of cerebrovascular damage in neurodegenerative diseases. In this review, we will discuss recent developments in our understanding of the mechanisms linking NOX activity, OS and neurodegenerative diseases, with particular focus on the neurovascular component of these conditions. We conclude highlighting current challenges and future opportunities to combat age-related neurodegenerative disorders by targeting NOXs.

## 1. Nicotinamide Adenine Dinucleotide Phosphate (NADPH) Oxidases (NOXs)

The nicotinamide adenine dinucleotide phosphate (NADPH) oxidases (NOX) enzymes are multi-subunit protein complexes. They are membrane-bound proteins and their main function is to transfer electrons across the plasma membrane to molecular oxygen—which results in the generation of the superoxide anion and subsequently reactive oxygen species (ROS), including hydrogen peroxide (H_2_O_2_) and hydroxyl radicals (OH^●^) [1,2,3,4].

NOX2 was the first NOX enzymes to be discovered and initially named g92^phox^ while studying respiratory bursts in neutrophils. Shortly after the discovery of NOX2, other members of NOX family were cloned [1,5,6,7,8,9]. In accordance with the most recent terminology, the NOX family comprises of NOX2, NOX1, NOX3, NOX4, NOX5, DUOX1 and DUOX2 isoforms. All seven NOX isoforms share not only conserved functions but also conserved structural properties (for a review see: References [3,10,11]). All NOX isoforms have at least six transmembrane domains, a flavin adenine dinucleotide (FAD) and NADPH-binding cytosolic domains [7,10,12,13].

Although ROS generation is the main function of the NOX enzymes, they vary in the way the enzymes are activated and the type of ROS generated. Native NOX proteins are inactive as a naïve monomer and are dependent on interacting proteins for their maturation, stabilisation, haem incorporation and their translocation to the membrane/site of activity [10,14]. NOXs 1–3 interact with p22^phox^ transmembrane protein along with the cytosolic organiser subunits (p47^phox^, NOXO1), activator subunits (p67^phox^/NOXA2, NOXA1, p40^phox^) and the G-protein Rac [15]. Conversely, NOX4 only requires p22^phox^ for activity and is thought be constitutively active, while NOX5, DUOX1 and DUOX2 activation depends on direct Ca^2+^ binding [3,10].

Though NOX enzymes are ubiquitously expressed, they have unique distribution patterns and expression levels in different tissues throughout the body [10,16]. For instance, NOX1 is highly present in the colon, NOX2 in phagocytes, NOX3 in the inner ear, NOX4 in the kidney and blood vessels, NOX5 in the lymphoid tissue and the testis, DUOX1 and 2 in the thyroid [3,7,12,14].

All NOX isoforms act as a catalyst for the transfer of two electrons from NADPH through their FAD domain and two haem prosthetic groups to molecular oxygen [17]. NOXs 1, 2, 3 and 5 generate superoxide, while NOX4, DUOX 1 and 2 largely release hydrogen peroxide [18,19]. ROS production by activated NADPH oxidase is triggered by a wide variety of factors including mechanical forces, environmental factors (e.g., hypoxia) and cytokines and hormones such as angiotensin II (AngII), aldosterone, endothelin-1 (ET-1), platelet-derived growth factor (PDGF), TGFβ (transforming growth factor β) and TNFα (tumour necrosis factor α) [20,21,22,23].

NOX enzymes play a significant role in several biological processes such as host defence, cellular signalling, metabolism, stress response, transcription and translational regulation and tissue homeostasis [3,4,24]. In this review, we will discuss the role of oxidative stress in central nervous system (CNS) degeneration, the developments in understanding the role of NOX enzymes in neurovascular disorders and the current challenges in targeting NOXs pharmacologically.

## 2. Oxidative Stress: NOX-ROS

Within a cell, ROS is produced in multiple cellular compartments including the cytosol, peroxisomes, endoplasmic reticulum; however almost 90% of the ROS is generated and compartmentalized in the mitochondria [25]. Mitochondria produce ROS (mtROS) as byproducts of the ATP generation by the oxidative phosphorylation (oxphos). Superoxide anion can in fact be released by the electron transport chain complexes I and III [26], which is rapidly dismutated to H_2_O_2_ by mitochondrial SOD2. Thus, historically, ROS was assumed to be produced accidently in the cells as a harmful by-product of aerobic metabolism. However, over the last decade ROS have been intensively investigated as mediators of both physiological and pathophysiological signals [27,28,29]. Mitochondrial damage and the resulting oxidative stress results in the formation hydroxyl radical (^•^OH), the most potent and destructive oxidant via the Fenton reaction (for review see: References [25,26,30,31,32,33]). Although mtROS has been implicated in various neurological disorders, however, for the purpose of this review we will be focusing on ROS generation by the NADPH enzymes.

Free radicals contain one or more unpaired electrons in their outer shell and are chemical intermediates in the reduction of molecular oxygen to water [34,35] as per the following reaction:
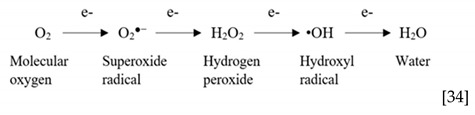


The initial electron transfer to oxygen by NOXs and other cellular enzymes results in the generation of the superoxide radical O_2_^●−^ which is dismutated (spontaneously or by SODs) to H_2_O_2_ by addition of an electron and 2H^+^. Even though H_2_O_2_ does not have an unpaired electron and is not a free radical, it is an effective oxidant for a number of biological molecules. In the cellular microenvironment, it is the reduction of H_2_O_2_ that yields the hydroxyl radical (^•^OH), the most reactive oxygen radical which is capable of oxidizing lipids, carbohydrates, protein and DNA [35]. O_2_^●−^ is short lived, has limited diffusibility across membranes compared to H_2_O_2_, which is more stable and membrane diffusible. Highly reactive peroxynitrite (ONOO^−^) is formed when superoxide reacts with nitric oxide (NO). Other species include hypochlorous acid (HOCl) and singlet oxygen (^1^O_2_) which are generated when peroxidase enzymes catalyse reactions involving H_2_O_2_ [36].

Low levels of cellular ROS have been shown to be important in normal biological functions, however excessive levels of ROS and the resulting oxidative stress that causes biological perturbations [37]. Proteins are major cell targets of free radicals within the cell [38]. ROS can modify proteins by fragmenting the peptide chain, altering the electric charge, cross-linking of proteins and addition of carbonyl groups—all of which make the proteins susceptible to proteolytic degradation [39,40,41,42]. In particular, cysteine and methionine residues and their sulfhydryl groups are more vulnerable to oxidation resulting in conformational changes, protein unfolding, and degradation [40]. The oxidative damage to proteins eventually leads to inhibition of enzymatic/binding activities, increased susceptibility to aggregation and proteolysis, altered uptake by cells, and altered immunogenicity [43]. Several transcription factors (TFs) have been shown to be activated by ROS and amongst redox-regulated TFs, important examples are Forkhead homebox type O (FOXO) [44,45], nuclear factor E2-related factor 2 (Nrf2) [46,47], tumour suppressor 53 (p53) [48], nuclear factor kappa B (NF-κB) [49,50,51], activator protein-1 (AP-1) [52,53] and hypoxia inducible factor 1-a (HIF-1a) [54]. FOXO TFs can be viewed as molecular sensors for oxidative stress since their activity is regulated by H_2_O_2_. Once activated, they relay the stress signals to induce apoptosis, stress resistance, or senescence [55,56]. Activation of NF-κB via oxidation-induced degradation of the inhibitory subunit IκB results in the activation of several antioxidant defence-related genes such as *Nrf2, HO-1* [57]. NF-κB also regulates the expression of genes regulating immune responses, such as IL-1β, IL-6, tumour necrosis factor-α (TNF-α), IL-8, and several adhesion molecules [50,58]. ROS also induces lipid peroxidation which disrupts the membrane lipid bilayer arrangement which has been shown to inactivate membrane-bound receptors and enzymes resulting in increased tissue permeability [59,60]. Oxidative modifications of the DNA include degradation of bases, breaks in the DNA strands, mutations, translocations and abnormal cross-linking with proteins [40].

## 3. ROS and Aging in the Central Nervous System (CNS)

The aging process is characterised by a gradual, cumulative deterioration in physiological functions over time which results in increased susceptibility to disease [61,62]. During the mid-1950s Denham Harman proposed the ‘free radical theory’ which is now the most acceptable explanation for the mechanistic basis of aging [61,63]. According to this theory, accumulation of free radicals over time induce deleterious damage to cellular macromolecules and the inability to counter these changes by anti-oxidant mechanisms results in aging and age-related diseases [61,64]. Aging occurs in all organs and the systems affected by aging include the immune system, the cardio vascular system and the CNS [62].

The brain consumes about 20% of the bodily oxygen and yet, has relatively weak antioxidant mechanisms [65]. This makes the brain extremely susceptible to ROS insults. As such, the brain also undergoes structural and functional changes with aging. The brain mass decreases in the order of 2% to 3% every decade for individuals over 50 years of age. At the cellular levels, ageing is accompanied by telomere shortening, DNA damage, oxidative stress and inflammation at the cellular level [66]. Studies in human and mice brains have reported that the anti-inflammatory cytokine IL-10 decreases and pro-inflammatory cytokines such as TGF-β, TNF-α and IL-1β increase in the CNS with age [67,68,69]. Age-related changes in the brain microenvironment eventually lead to inflammation, increased permeability of the blood brain barrier (BBB), neuronal degeneration and ROS production [70].

Microglia and astrocytes are considered to be the main mediators of age related neuro-inflammation [66] and studies have reported that increase in microglial activity in the brain could be an early event that leads to oxidative damage [66,71,72,73,74]. Astrocytes and neurons in the hippocampus secrete TGFβ, which regulates microglial activity [66,75,76]. TGF-β is the key regulator of neuro-inflammation as it controls the switch of microglia from protective to deleterious [66]. Microglial cells are the CNS resident macrophages and provide the first line of defence in the brain [61,77]. When activated, microglial cells increase in size, number and release superoxide and nitric oxide which have been shown to cause neuronal cell death and oxidative damage under pathological conditions [61]. Microglia from aged brains show upregulated IL-6 and IL-1β compared to young brains in mice [66,73]. Increased levels of TGFβ with age also activate other inflammatory pathways, such as ERK/MAPK, P38, MAPK, JNK and the PI3K signalling which result in further microglial activation, oxidative stress and neuroinflammation [78]. Aged microglia undergo a decay in lysosomal and mitochondrial activity which exaggerate ROS generation and the increase in ROS activates NF-κB thus exacerbating neuro-inflammation [72,79]. Reactive microgliosis seems to be a major event in CNS pathogenesis along with altered inflammatory cytokines [80,81] and augmented microglial NADPH-derived ROS accumulation [82]. Subsequent pathophysiological events leading to neurodegeneration lead to loss of the integrity of the blood brain barrier (BBB), and increased release of inflammatory cytokines into the cerebral spinal fluid (CSF) [80,81,83].

## 4. Dementia

Dementia is an umbrella term for a set of symptoms that are progressive and debilitating, leading to loss of memory, cognition impairment and behavioural changes affecting the individual’s quality of life. Longer life expectancy, along with the decline of infectious diseases globally is associated with a steep increase in the incidences of dementia [84]. Currently, there are about 850,000 cases of dementia in the UK alone, with this number set to reach a million by 2025. As a result, dementia, the most common neurodegenerative disorder, is rapidly becoming the most serious medical problem of the current century [84,85,86]. Dementia is categorised as follows: (i) Degenerative dementias: (ia) With cortical features (Alzheimer’s disease (AD) and Frontotemporal dementia (FTD)); (ib) With subcortical features (Parkinson’s disease, Huntington’s disease); (ic) With both cortical and sub-cortical features (Lewy body); (ii) Vascular dementia; (iii) Mixed dementia (both degenerative and vascular); and (v) dementia due to trauma or other medical conditions [87]. The most common types of dementia are discussed below:

### 4.1. Degenerative Dementias

#### 4.1.1. Cortical dementia

*Alzheimer’s disease (AD)*: Alzheimer’s disease (AD) is the most common form of dementia, the cause of which still largely remains unknown. AD accounts for nearly 50% to 75% of all dementias and according to the world Alzheimer report, 46.8 million individuals were suffering from dementia in 2015, with this number expecting to double every 20 years [83]. AD results in loss of cognition, memory and cerebral tissue functions that eventually cause death [88]. The extracellular accumulation of amyloid β (Aβ) peptides (Aβ plaques) in the senile plaques (SP) and intracellular deposition of hyperphosphorylated tau protein (neurofibrillary tangles (NFT)) in the hippocampus and neocortex form the hallmarks of this disease [29,89,90]. Aβ deposits are also observed in the cerebrovasculature resulting in the development of cerebral amyloid angiopathy (CAA) [91]. CAA is the major cause of white matter hyperintensity, cerebral microbleeds and intracerebral haemorrhage, loss of cognition in the elderly. 

AD can be classified genetically; familial cases with Mendelian inheritance (Familial AD (FAD)) and sporadic AD. Almost 90% of cases of AD are sporadic with individuals affected at ages older than 65. Familial cases of AD are predominantly early-onset (EOAD), representing 6% to 10% of all AD cases and the individuals affected are between 30 and 65 years of age [92]. Rare mutations discovered in three genes are responsible for EOAD where individuals die within 5 to 8 years after developing symptoms of AD [92,93]. Point mutations in β-APP (Amyloid precursor protein) located on chromosome 21 is sufficient to cause dominant, autosomal AD with complete penetrance. Individuals with this mutation show similar phenotypes as a typical AD patient. The type of mutation determines the length of the β-amyloid protein after cleavage by γ- and β-secretase of the APP; the longest form (42 amino acids) aggregates more readily than the short form (40 amino acids) [94]. In addition, mutations in Psen1 (Presenilin) 1 and 2 are also identified with cases of FAD. Psen1 and β-APP interaction in the CNS is crucial for systematising vesicular traffic. Interruption of this process leads to neuronal loss, excess β-APP metabolism which eventually leads to the formation of the long form of β-amyloid (Aβ1-42) [29,92,94].

The sporadic form of AD is more complex and most likely consequences from a combination of genetic and environmental factors. A genetic polymorphism that is a major risk factor for AD is the apolipoprotein E (APOE) gene. APOE is important for synaptic repair and maintenance of neuronal structure and APOE expression as the ε4 allele increases the risk of developing AD up to 15-fold compared to the others (ε2 or 3). Individuals with this genotype have a 95% chance of developing AD by the age of 80 [93]. More recently, two other genetic factors have been linked with AD development; the HLA-A2 allele is associated with early onset AD; and changes in the mitochondrial genome contributing to oxidative stress plays a key role in the progression of AD [93]. In addition to the genetic alterations, aging and associated risk factors such as hypertension, diabetes, smoking and obesity remain to be the multifactorial causes of AD. Interestingly, genetic changes, metabolic defects and aging are also associated with the production of high levels of ROS and a number of studies are converging on the fact that oxidative stress might be a crucial player in driving AD pathogenesis. Aβ is associated with increased levels of oxidation in cells of the hippocampus and cortex of the AD brain [94]. When compared to healthy controls, AD patients presented upregulated levels of inflammatory cytokines and chemokines including IL-1β, IL-18, TNF-α, TGF-β, MIP1-α and CCR5 [80,81,83] suggesting a role for amyloid β in neuro-vascular inflammation.

*Frontotemporal dementia (FTD)*: FTD consists of a collection of syndromes characterised by degeneration of the frontal and the temporal lobes. Two major types of FTD have been described, which depend on the symptoms: Behavioural and language type. FTD is the second most frequent form of dementia observed in younger individuals with a prevalence of 15 to 22 every 100,000 people between 45 and 65 years of age. FTD is familial in nearly 30% to 50% of the cases and is characterised by disinhibition, socially unsuitable manners, loss of empathy and hyperorality [95]. Mutations in five unrelated genes (Tau/MAPT, c9orf72, GRN, CHMP2B and VCP) have been identified as a cause for most FTD but not all cases of familial FTD. All individuals with FTD present with neuronal loss and astrocytic gliosis but the major protein involved differs. The proteins that undergo abnormal processing and deposition are Tau (Pick’s disease), transactive DNA response binding protein (TDP) and tumour associated protein fused in sarcoma (FUS) [95]. 

#### 4.1.2. Subcortical Dementia

This type of dementia is associated with irregularities within the cerebral cortex, either by disorders of cortical afferences or efferences or an insufficient supply in neuromodulators [96]. This type of dementia is characterised by symptoms such as forgetfulness, slowness of thought, emotional changes, changes in personality and inability to process knowledge [96].

*Parkinson’s disease dementia (PDD)*: PD is caused by loss of nerve cells in the substancia nigra in the brain. As a result of the loss of these nerves in this part of the brain, the production of dopamine is severely reduced. Dopamine is a chemical that acts as a second messenger, co-ordinating control and bodily movements. PDD is a cognitive impairment that appears in the condition of pre-existing PD [97]. PD affects about 1 in 500 individuals and most start developing symptoms over 50 years of age. The clinical symptoms of PDD include defects in memory, attention and visual perception that worsen with disease progression [98,99]. Although most PD have an unknown cause, mutations in LRRK2, PARK7, PINK1, PRKN, or SNCA genes are the risk factors in cases of familial PD.

*Huntington’s disease (HD)*: HD is a progressive degeneration of the brain which results in uncontrolled movements, loss of emotion and cognition that affects 3–7 every 100,000 people of European descent [100]. Adult onset HD is the most common form and individuals live up to 15–20 years after the initial symptoms begin. Juvenile HD is less common and begins in childhood or adolescence. Juvenile HD progresses rapidly, and affected individuals live up to 15 years of age after initial diagnosis. The disruption in the translation of huntingtin protein due to mutations in the HTT gene causes HD. Mutated HTT is autosomal dominant and affected individuals inherit the gene from one affected parent [100,101].

*Lewy body dementia (LBD)*: LBD is an atypical PD, with the characteristic finding of lewy bodies (LB) and accounts for nearly 25% of all dementias. It is prevalent in 0.7% of the population, over 65 years of age. LB is a neuronal inclusion comprised of aggregated α-synuclein found in the brain stem nuclei and loss of cholinergic neurons. Clinical symptoms of LBD include loss of alertness, hallucinations and loss of motor functions. Studies have shown that mutations in GBA1 gene and CYP2D68 are associated with LBD onset, however there are no post-mortem tests to confirm LBD diagnosis [102].

### 4.2. Vascular Dementia (VaD)

VaD is the second most common type of dementia and the main cause is reduced blood flow to the brain. The prevalence of VaD increases steeply with age and the mortality of patients with VaD is higher than that of AD patients. Atherosclerotic diseases, microbleeds and strokes are in general considered the major risk factors for VaD. Brain lesions related to vascular defects are heterogenous, which leads to different cognitive defects such as speech disorders, seizures, white matter lesions, abnormalities in gait, incontinence and Parkinsonism [103,104]. The clinical diagnosis of VaD is quite straightforward which includes neuroimaging and neuropathological examinations. The criteria for VaD assessment are derived from expert opinions based on current knowledge and pathogenetic hypothesis of dementia [105]. The classification of VaD depends on different factors, which include number, age, origin and type of vascular lesions, presence of haemorrhage, distribution of arterial territories, anatomical localisation and size of vessels involved [106]. The severity of VaD ranges from mild cognitive impairment (MCI) to severe dysfunction [96]. Vessel damage due to atherosclerosis or small vessel disease can lead to thrombus formation causing vessel occlusion, microaneurysms and necrosis of the vessel wall [106,107].

### 4.3. Mixed Dementia

Mixed dementia (MD) is prevalent in about 45% of cases where neurodegenerative and vascular processes are mutually potentiated. In such a scenario, the individual has characteristics of both AD and VaD [108]. The small vessels within the brain play a key role in the clearance of Aβ peptides as sites of efflux across the BBB and as sites of perivascular drainage. In AD, Aβ deposits are observed on the vessel walls which result in vascular fragility and vessel wall necrosis [109]. This degenerative vascular condition is termed cerebral amyloid angiopathy (CAA) [91,110]. CAA is the most common cause of intracerebral haemorrhage (ICH) and is often characterised by ischemic lesions, micro- and macro-haemorrhages and impaired cerebral blood flow. A number of recent studies support that CAA in turn also exacerbates the failure in the elimination process of Aβ and ultimately, impaired blood flow and resulting ischemia aggravates neurodegeneration and accelerates the progression of AD. Aging is the strongest risk factor for CAA and is most common in the elderly. Population based autopsies indicate the presence of CAA in 20% to 40% in non-demented and 50% to 60% in demented elderly (80–90 years) individuals. Almost all AD brains have CAA with up to 25% of AD brains presenting advanced CAA [107]. It is striking that although amyloid β deposition is the main cause of CAA development, less than 50% of CAA cases meet the pathologic criteria of AD or over 75% patients with AD have mild to no CAA [91].

As is the case with most dementias, the pathological onset of MD is several years prior to the display of symptoms [111]. Cognitive performance of patients with MD is lower than that of AD, particularly in memory, denomination and executive functions.

## 5. Oxidative Stress and Dementia

Most eukaryotic organisms require oxygen for energy metabolism for normal biological functions. Cell metabolism generates ROS/RNS, which at physiological concentrations regulate a wide array of functions such as cellular signalling, host defence, induction of mitogenic responses and cerebro-vascular perfusions [112,113]. The neuro-vascular unit (NVU) consisting of neurons, astrocytes, pericytes, microglia and endothelial cells are capable of anti-oxidant defence mechanisms (glutathione (GSH), glutathione peroxidase, glutathione reductase, SOD and catalase) that not only maintain physiological levels of ROS but also maintain the integrity of the BBB [113].

However, a failure in the ability of NVU cells to maintain the proper balance between ROS production and their neutralization causes the disruption of brain homeostasis resulting in oxidative stress (OS) [112]. The CNS is predominantly vulnerable to OS due to its high oxygen consumption and the abundance of easily oxidized polyunsaturated fatty acids, putting OS at the core of many neurodegenerative diseases [65]. Increased ROS formation plays a major role in the alteration of NVU function [114,115]. It has been reported that OS in the NVU can lead to an increase in BBB permeability resulting in the entry of neurotoxic substances into the brain, impair the CNS nutrient delivery system and eventually lead to neuronal loss and synaptic dysfunction [113,116]. OS and inflammatory conditions within the brain cause demyelination and axonal damage [65], which further results in the generation of peroxynitrate molecules that cause severe damage to neurons [117]. In fact, OS is shown to be the key mediator in pathogenesis of PD, AD and stroke [118]. In PD, oxidative stress plays a key role in dopaminergic neurotoxicity. Activated microglia release free radicals which causes unwarranted and uncontrolled neuro-inflammatory responses, leading to a self-perpetuating, vicious cycle of neurodegeneration [119]. The oxidised molecules such as neuromelanin, α-synuclein and activated MMP3 released from the damaged dopaminergic neurons further trigger the activation of microglial cells. Studies have shown that MMP3 KO mice not only have lower levels of superoxide production, but also show abrogated microglial activation following exposure to MPTP compared to WT mice [120].

Aβ accumulation is associated with increased levels of oxidation in cells of the hippocampus and cortex of the AD brain [94]. Aβ also facilitates expression of the receptor for advanced glycation end products (RAGE) in microvascular endothelial cells neurons and microglia [34]. RAGE is a multiligand receptor which is able to bind to advanced glycation end-products (AGEs), amphoterin, calgranulins, and Aβ peptides. RAGE expression induces ROS mainly through the activity of NADPH oxidases [121]. Indeed, increased RAGE expression is associated with deposition of Aβ in the cerebral intracellular space which exacerbates AD-associated neuronal damage [29,122]. Free radicals have been shown to play a significant role in oxidation of brain macromolecules [34]. Lipid peroxidation increases in AD and the poly unsaturated fatty acids (PUFA) which make up the brain’s membrane phospholipids become vulnerable to oxidative insult. PUFA oxidation results in the generation of multiple aldehydes such as propanal, butanal, pentanal and 4-hydroxynonenal (HNE). HNE is extremely reactive and together with OS inhibits protein synthesis, glycolysis and causes DNA/RNA and protein degradation. These alterations are closely related to oxidative imbalances that result in BBB permeability [113]. In the AD individuals, HNE-protein adducts are present in the NFT and is significantly associated with the APOE ε4 carriers of AD [34,113]. Interestingly, Lourenco et al. (2017) showed that in the 3xTg-AD mouse model of AD, the cerebral blood flow was impaired prior to memory dysfunction, suggesting that cerebro-vascular alterations, CAA could be the driver of AD pathogenesis [123].

## 6. NOX and Dementia

Due to the high aerobic activity in the brain, mitochondria were considered to be the most likely source of ROS. Nevertheless, Trifunovic et al., using a mutated mitochondrial DNA (mtDNA) mice showed that this was not the case. Mice with mtDNA mutator phenotype were prone to increase in mutations and deletions in the mtDNA which correlated with the onset of pre-mature aging related phenotype. However, there was no change in the amount of ROS produced nor damage from OS observed [124]. These results suggested that there must be another source of ROS, upstream of the mitochondria [125]. The NOX enzymes are activated in the microglia and are also detected in the astrocytes and neurons [66,125]. The NOX mRNA transcripts have been detected in total brain samples and NOX 1, 2 and 4 have been shown to be expressed in the different areas of the CNS including the neurons, astrocytes, microglia and the cerebro-vascular cells forming the blood brain barrier [126,127,128,129,130]. The activity of NOXs has been shown to be higher in cerebral arteries compared to the peripheral arteries and, high levels of NOX2 have been detected in the endothelial and adventitial cells of the cerebral arteries, with NOX2 expression being the highest, followed by NOX1 and NOX4 [131]. NOX and Rac1 activation in the hippocampus has been associated with cognitive damage following cerebral ischemia [132,133]. Moreover, NOX activation is also associated with hippocampal cell death in rats with cerebral hyperfusion [131,134,135].

It has been reported that following injury, activated microglia and astrocytes produce high levels of ROS via NOXs, which have deleterious effects in the expression of important molecules involved in BBB integrity (e.g., ZO-1, claudin-5 and occludin). Additionally, under pathological conditions, pericytes (cells in close proximity to the endothelial cells) are highly susceptible to OS and undergo apoptosis thus further weakening the BBB [71,136].

In an in vivo model of PD (i.e., systemic administration or striatal injection of 1-methyl-4-phenyl-1,2,3,6-tetrahydro-pyridine (MPTP)) translocation of p67phox was induced and p47phox phosphorylation and p47phox–gp91phox complexes were also significantly increased in mice substantia nigra (SN) [137]. It has also been shown that gp91phox co-localises with microglial cells in the MPTP-induced PD mouse model, suggesting a role for microglial cells as a key site of NOX activation [138].

Seminal work by Ansari and Scheff (2011) systematically compared individuals with pre-clinical AD (PCAD), mild cognitive impairment (MCI) and early-moderate AD (mAD) at ante-mortem cognitive testing and post-mortem histopathology. They found that NOX activity was most elevated in the MCI cohort and remained elevated in all AD cohorts compared to NCI. Additionally, not only were NOX cytosolic subunits significantly elevated compared to the membrane bound subunits but with increased NOX activity cognitive performance of the individual also decreased [16]. Activation of NOX2 in the brain of AD subjects has been demonstrated and confirmed by the translocation of NOX2 subunits to the cell membrane [139]. In addition, NOX1 and NOX3 mRNA has also been detected in the frontal lobe of AD brains compared to normally aged brains [140] suggesting a role for multiple NOX isoforms in neuronal degeneration. NOX activity in vitro can be increased by Aβ peptides, similarly there is a direct correlation with age dependent Aβ deposition and NOX activity in the humanised APPxPS1 knock-in mouse model [141]. Studies have shown that exposure of hippocampal neuronal-microglial co-cultures with Aβ peptides resulted in NOX activation, which resulted in neuronal cell death [142]. In another study, APP over expression in microglial and neuro-blastoma cells resulted in upregulated ROS generation which was repressed by ROS scavengers and inhibited in a dose-dependent manner by the NOX inhibitor DPI [142]. Interestingly, deletion of gp91^phox^ was also recently shown to decrease neurovascular dysfunction and cognitive decline in transgenic mice overexpressing the Swedish mutation of human APP [143]. In AD, accumulation of Aβ results in the activation of RAGE. One of the main signalling pathways mediated by RAGE activation is NOX signalling, leading to ROS production and altered gene expression. Indeed, within the last decade several studies have directly linked NOX upregulation to neuronal death and cerebrovascular damage [3,36,131,142,144,145]. RAGE activation also upregulates RAS-dependent signalling, activating MAP kinases (JUN, ERK1/2 or p38) and JAK/STAT signalling, all of which result in protein aggregation and production of pro-inflammatory molecules. In addition, it has been shown that RAGE facilitates Aβ transport via endocytosis and transcytosis across the BBB, thus promoting pathological accumulation of Aβ in the brain parenchyma [121]. Interestingly, RAGE also acts as a carrier for Aβ on neuronal cell surface and RAGE-dependent p38 MAPK activation promotes the internalization of the whole Aβ-RAGE complex into the cytosolic compartment, leading to mitochondrial dysfunctions, oxidative stress, and neuronal damage. These injuries are further deteriorated by Aβ deposition resulting in a feedback loop where AD associated cerebrovascular abnormalities cause hyperfusion, hypoxia and inflammation compromising the BBB permeability which results in failure of Aβ clearance ultimately leading to neurodegeneration and cognitive decline [146,147] (Figure 1).

## 7. NOXs as Therapeutic Targets

As ROS are recognized causes of neurodegeneration, there are clinical trials that have, and are investigating the role of anti-oxidants as dietary supplements in cognitive impairment. A trial for N Acetylcysteine (NAC) has shown promise in vascular cognitive impairment-no dementia (VCI-ND) patients and is in phase 2 of clinical trial (Clinical trials reference: NCT03306979). Another trial is now recruiting patients to study the safety and efficacy of anthocyanins in improving dementia and related symptoms in older people (Clinical trials reference: NCT03419039). Anthocyanins, belonging to the flavonoids class of molecules have previously been modulate NOX activity and reduce superoxide production in human vascular cell models [148].

Oxidative stress due to the NOX enzymes is now believed to be a key mediator of neuroinflammation and neurodegeneration. In fact, in the last decade, the main research focus of several groups has been to delineate the role of NOXs in CNS and CNS-related disorders, and as a result NOXs have emerged to be promising therapeutic targets.

However, the lack of isoform specific NOX inhibitors and exclusive NOX-derived ROS inhibitors remain a major challenge for the field. Indeed, several studies reported striking benefits of NOX inhibition using the natural product apocynin in alleviating symptoms of neuro-vascular diseases by decreasing microglial activation and improving survival in animal models [125]. However, further studies revealed that although at high concentrations, apocynin inhibits NOXs, it also acts a mitochondrial ROS scavenger. Thus, studies detailing therapeutic advantages with apocynin do not exclusively apply to NOXs.

Diphenyleneiodonium (DPI) irreversibly binds to the cytosolic FAD domain and inhibits all NOXs with high affinity. In subpicomolar concentrations, DPI displayed no toxicity to neuron-glia cultures in vitro and successfully inhibited NOX2 [149]. Yet, DPI inhibits other electron transport chains (NO synthase, xanthine oxidase), has low solubility and mitochondrial toxicity.

Since then, numerous small molecule NOX inhibitors have been developed using drug discovery approaches which include GKT137831, ML171 (2-APT), and VAS2870 which show not only improved specificity but also isoform selectivity [17] (Table 1). The GKT compounds: GKT136901 and GKT137831were developed by GenKyoTex. These compounds inhibit NOX1, 4 and 5 at lower concentrations and inhibit NOX2 at higher concentrations. Interestingly, GTK137831 has been successfully tested in vivo against liver fibrosis and diabetic complications [150,151,152]. More recently, a new compound GKT771 has been developed as a more potent and highly selective inhibitor for NOX1. GKT771 has been selected as a preclinical development candidate and is currently in phase I of clinical studies [153]. While still therapeutically relevant, the lack of complete clinical data and non-specificity of the older GKT compounds make them difficult to construe in context to NOXs.

Gianni et al., at the Scripps research institute showed ML171 to be successful in inhibition of OS in colon cancer cells. ML171 or 2-(acetyl)-phenothiazine (2-APT) was identified as a potent NOX1 inhibitor at nanomolar concentrations, and a NOX2, 4 and xanthine oxidase inhibitor at higher concentration [155]. Additionally, 2-APT also attenuates NOX1 mediated GPVI signalling and ROS production, however, despite of its higher specificity for NOX1 it needs to be further tested in in vivo studies for mechanisms and drug safety [17,156].

The VAS compounds (VAS2870, VAS3947) were developed by Vasopharm GmbH in a screening approach to develop NOX2 inhibitors. VAS showed specificity towards inhibition of NOX1, 2, 4 and 5 in a concentration dependent manner and did not affect xanthine synthase or eNOS pathways. However, the use of this drug is limited by poor solubility, lack of pharmacokinetic studies and recent data by Sun et al., showing that VAS2870 exerts significant off-target effects by alkylating cysteine residues in receptors for Ca^2+^ channels [157].

Other inhibitors developed include S17834, Fulvene-5, Triphenylmethane derivatives, Shinogi I and II, Diarylheptanoids, Ebselenm, Celastrol and Perhexiline. Most of these compounds either lack specificity for NOXs, characterisation of isoform selectivity or lack in mechanisms of action thus, limiting their use in therapeutically applicable studies.

Another approach is to use biologicals (regulating peptides or antibodies) as drugs. The NOX-derived peptide most commonly used is the NOX2 docking sequence (NOX2ds)-tat (Tat is viral replication protein that has the ability to translocate through the plasma membrane and enter the nucleus to transactivate the genome [158]). NOX2ds-tat, 18-amino acid long peptide was the first biological NOX inhibitor to be designed. This peptide contains a 9-amino acid sequence of the intracellular B-loop of NOX2 that binds to p47^phox^ and thus preventing the NOX2 complex assembly [159,160]. NOX2ds-tat has shown selectivity for NOX2, albeit in cell-free assays. The peptide has been used in in vivo models and has shown to be effective in superoxide inhibition and protection from neurovascular dysfunction in mice over-expressing APP and hypertension compared to the scrambled peptide [143,161]. Another peptide inhibitor, NOXA1ds contains 11 amino acid sequences of NOXA1 to NOX1 [162]. NOXA1ds is selective for NOX1 however, its mechanisms of actions in living systems still remains to be characterised. Another major limitation that restricts the peptides for use in therapeutics is the potential antigenic responses and hence, further studies will be needed to be carried out to establish the long-term effects of these peptides, their in vivo stability and NOX selectivity [17].

The initial promise and purpose of the small molecule and peptide inhibitors has grown over time and as the NOX field widens, it is expected to develop further.

## 8. Concluding Remarks

Recent studies have identified NOX isoform expression within the CNS and it is now undoubtedly accepted that the ROS-generating NOX enzymes play a significant role in the pathogenesis of several neurodegenerative diseases and neuroinflammation. Despite the significant scientific progress in understanding dementia and identifying OS as a common pathological feature in most dementia, there are no current therapeutics available to halt the neurodegenerative processes. In such a scenario, it might be noteworthy to consider that NOX activation, increased ROS generation and the resulting OS can be driving or at least contributing to the neurodegenerative process responsible for dementia. The fact that disease progression shows a direct correlation with NOX activity makes the NOXs enzymes extremely promising biomarkers when assessing and developing novel therapies. Corroboratory studies defining the mechanism of action and role of the individual NOX isoforms in homeostatic and disease conditions within the CNS will need to be carried out. Our increased understanding of the mechanism of action of NOXs and the structural biology studies of NOXs and their regulatory subunits are paving the way for the development of potent and specific inhibitors. The development of such novel molecules will serve as a basis for development of original and promising therapies for pathological conditions associated with NOX activation, which include dementia.

## Figures and Tables

**Figure 1 ijms-19-03824-f001:**
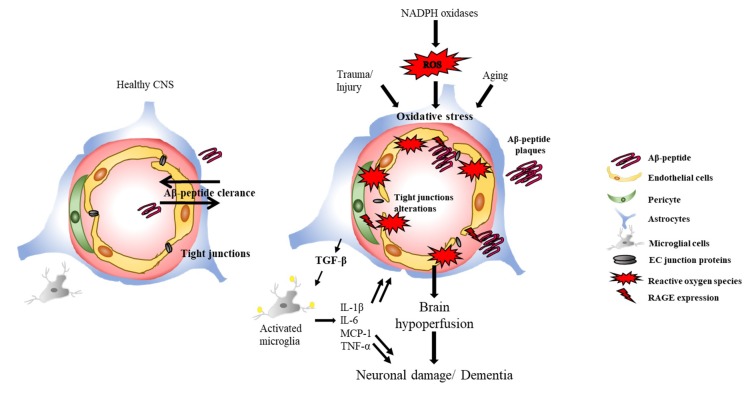
Oxidative stress and blood brain barrier (BBB) breakdown: Reactive oxygen species (ROS) accumulation as a result of NADPH oxidases (NOX) activation, trauma or aging results in the accumulation of Aβ-protein, inflammatory and immune responses. Aβ accumulation also results in cerebral amyloid angiopathy (CAA) that leads to the failure in Aβ clearance. Aβ deposits on the vessel walls result in vascular fragility, loss of endothelial junction proteins and microglial activation. Aβ also facilitates the expression the receptor for advanced glycation end products (RAGE) in microvascular endothelial cells, neurons and microglia. Activated microglial cells release superoxides and nitric oxides which have been shown to cause neuronal cell death and oxidative damage under pathological conditions which eventually result in breakdown of the BBB, neuronal degradation and dementia.

**Table 1 ijms-19-03824-t001:** Mechanistic activity and specificity validation of selected catalytic subunits of nicotinamide adenine dinucleotide phosphate oxidase inhibitors [17,154,155]. XO: Xanthine Oxidase, eNOS: endothelial nitric oxide synthase, GPCR: G-protein coupled receptor.

Compound	Direct Interaction with NOX Complex	Off Target Effects
GKT136901	Yes (Cell free assays and in vivo)	Yes (XO)
GKT137831	Yes (Cell free assays and in vivo)	Yes (XO, eNOS)
ML171	*n.a.*	Yes, minor (5-HT2b receptor)
VAS2870	Inhibits NOX activity when added before the complex assembly	XO
VAS3947	Yes (Cell free assays)	
Celastrol	Yes (Cell free assays)	XO
Ebselen	Inhibits NOX activity when added before the complex assembly	XO
Perhexiline	Yes (Cell free assays)	
ACD084	Yes (Cell free assays)	Mitochondrial complex I
NOX2ds-tat	Inhibits NOX activity when added before the complex assembly	XO
NOXA1ds	Inhibits NOX activity when added before the complex assembly	XO

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
