# Peer review of "The Role of NADPH Oxidases and Oxidative Stress in Neurodegenerative Disorders"

_ijms, 2018, doi:10.3390/ijms19123824_

Round 1

Reviewer 1 Report

This review article focuses on the role of NOX in OS that leads to various senile dementia. NOX, while an integral part of the cell's redox system, is not the focus of most studies in the area of oxidative stress research. By reviewing advances on NOX research, this is a very informative and timely article to readers in the oxidative stress field and those who are interested in aging and dementia. While I suggest to publish this work, there are a couple of minor points I hope the authors can address.

This article seems to suggest that OS caused by NOX is more detrimental to brain aging and neurodegeneration than those from dysfunctional mitochondrial OXPHOS. While this may be the case, the OS state in aging cells may not matter whether it is caused by NOX or faulty mitochondrial molecular machinery. And the effects of antioxidant agents are often not as effective as previously expected. Is there a distinction between NOX-induced ROS and those produced by the mitochondrial OXPHOS system?

As evidenced by this article, NOX enzymes can serve as therapeutic targets for dementia and neurodegeneration. Besids their roles in ROS production, are there other possible mechanisms that might be at play here? This is partly connected to the question above. If there is not a clear distinction, that might suggest other possible mechanisms independent of their role in OS induction.

Author Response

This article seems to suggest that OS caused by NOX is more detrimental to brain aging and neurodegeneration than those from dysfunctional mitochondrial OXPHOS. While this may be the case, the OS state in aging cells may not matter whether it is caused by NOX or faulty mitochondrial molecular machinery. And the effects of antioxidant agents are often not as effective as previously expected. Is there a distinction between NOX-induced ROS and those produced by the mitochondrial OXPHOS system?

As evidenced by this article, NOX enzymes can serve as therapeutic targets for dementia and neurodegeneration. Besids their roles in ROS production, are there other possible mechanisms that might be at play here? This is partly connected to the question above. If there is not a clear distinction, that might suggest other possible mechanisms independent of their role in OS induction

In order to address this, we have now included a paragraph on mitochondrial ROS (Page 2, line 67). Although mitochondrial ROS (mtROS) and mitochondrial damage have been implicated in a number of neurodegerative diseases, for the purposes of this review, we have focussed on NOX- ROS and the mechanisms of CNS degeneration due to ROS imbalances.

Reviewer 2 Report

The authors have done an adequate job in compiling the recent literature for this manuscript. They have succinctly described the role of NADPH oxidases and OS in neurodegenerative disorders. However, I have a few suggestions to improve the manuscript further:
1) Line3: typo in the spelling of neurodegenerative.
2) discuss the Fenton reaction by which hydroxyl radicals are generated.
3)discuss the cleavage of the APP into sAPPalpha and sAPPbeta; it would be nice to have a diagram
4) Discuss how the electron transport chain is involved in the generation of OS and how leakage of radicals occurs.
5)Succinctly discuss ways to balance OS using antioxidants, etc.....

Author Response

1) Line3: typo in the spelling of neurodegenerative.

This has now been corrected.

2) discuss the Fenton reaction by which hydroxyl radicals are generated.

This has now been included in the manuscript under the section on mitochondrial ROS production (Page 2, line 67- 78).

3) discuss the cleavage of the APP into sAPPalpha and sAPPbeta; it would be nice to have a diagram

While this is extremely interesting, we fear that this has already been discussed by the authors in a previous publication (See: Canobbio I et. al., 2015. Frontiers in cellular neuroscience) and several other valuable literature reviews (Chow V.W. et. al., An Overview of APP Processing Enzymes and Products. Neuromolecular Med. 2010 Mar; 12(1): 1–12; O’Brien R.J. and Wong P.C. Amyloid Precursor Protein Processing and Alzheimer’s Disease. Annu Rev Neurosci. 2011; 34: 185–204; and Zhang Y. et. al., APP processing in Alzheimer's disease. Mol Brain. 2011; 4: 3).

4) Discuss how the electron transport chain is involved in the generation of OS and how leakage of radicals occurs.

We have now included a section on mitochondrial ROS production and OS (Page 2, line 67)

5) Succinctly discuss ways to balance OS using antioxidants, etc.....

This has now been added to the manuscript (Page 9, line 396).